# Consistent Multitask Learning with Nonlinear Output Relations

**Carlo Ciliberto** •,1    **Alessandro Rudi** •,*,2    **Lorenzo Rosasco** 3,4,5    **Massimiliano Pontil** 1,5

{c.ciliberto,m.pontil}@ucl.ac.uk    alessandro.rudi@inria.fr    lrosasco@mit.edu

[1]Department of Computer Science, University College London, London, UK.
[2]INRIA - Sierra Project-team and École Normale Supérieure, Paris, France.
[3]Massachusetts Institute of Technology, Cambridge, USA.
[4]Università degli studi di Genova, Genova, Italy.
[5]Istituto Italiano di Tecnologia, Genova, Italy.
• Equal Contribution

## Abstract

Key to multitask learning is exploiting the relationships between different tasks in order to improve prediction performance. Most previous methods have focused on the case where tasks relations can be modeled as linear operators and regularization approaches can be used successfully. However, in practice assuming the tasks to be linearly related is often restrictive, and allowing for nonlinear structures is a challenge. In this paper, we tackle this issue by casting the problem within the framework of structured prediction. Our main contribution is a novel algorithm for learning multiple tasks which are related by a system of nonlinear equations that their joint outputs need to satisfy. We show that our algorithm can be efficiently implemented and study its generalization properties, proving universal consistency and learning rates. Our theoretical analysis highlights the benefits of non-linear multitask learning over learning the tasks independently. Encouraging experimental results show the benefits of the proposed method in practice.

## 1   Introduction

Improving the efficiency of learning from human supervision is one of the great challenges in machine learning. Multitask learning is one of the key approaches in this sense and it is based on the assumption that different learning problems (i.e. tasks) are often related, a property that can be exploited to reduce the amount of data needed to learn each individual tasks and in particular to learn efficiently novel tasks (a.k.a. transfer learning, learning to learn [1]). Special cases of multitask learning include vector-valued regression and multi-category classification; applications are numerous, including classic ones in geophysics, recommender systems, co-kriging or collaborative filtering (see [2, 3, 4] and references therein). Diverse methods have been proposed to tackle this problem, for examples based on kernel methods [5], sparsity approaches [3] or neural networks [6]. Furthermore, recent theoretical results allowed to quantify the benefits of multitask learning from a generalization point view when considering specific methods [7, 8].

A common challenge for multitask learning approaches is the problem of incorporating prior assumptions on the task relatedness in the learning process. This can be done implicitly, as in neural networks [6], or explicitly, as done in regularization methods by designing suitable regularizers [5]. This latter approach is flexible enough to incorporate different notions of tasks' relatedness expressed, for example, in terms of clusters or a graph, see e.g. [9, 10]. Further, it can be extended to *learn* the tasks' structures when they are unknown [3, 11, 12, 13, 14, 15, 16]. However, most

regularization approaches are currently limited to imposing, or learning, tasks structures expressed by linear relations (see Sec. 5). For example an adjacency matrix in the case of graphs or a block matrix in the case of clusters. Clearly while such a restriction might make the problem more amenable to statistical and computational analysis, in practice it might be a severe limitation.

Encoding and exploiting *nonlinear* task relatedness is the problem we consider in this paper. Previous literature on the topic is scarce. Neural networks naturally allow to learn with nonlinear relations, however it is unclear whether such relations can be imposed a-priori. As explained below, our problem has some connections to that of manifold valued regression [17]. To our knowledge this is the first work addressing the problem of explicitly imposing nonlinear output relations among multiple tasks. Close to our perspective is [18], where however a different approach is proposed, implicitly enforcing a nonlinear structure on the problem by requiring the parameters of each task predictors to lie on a shared manifold in the hypotheses space.

Our main contribution is a novel method for learning multiple tasks which are *nonlinearly* related. We address this problem from the perspective of structured prediction (see [19, 20] and references therein) building upon ideas recently proposed in [21]. Specifically we look at multitask learning as the problem of learning a vector-valued function taking values in a prescribed set, which models tasks' interactions. We also discuss how to deal with possible violations of such a constraint set. We study the generalization properties of the proposed approach, proving universal consistency and learning rates. Our theoretical analysis allows also to identify specific training regimes in which multitask learning is clearly beneficial in contrast to learning all tasks independently.

## 2    Problem Formulation

Multitask learning (MTL) studies the problem of estimating multiple (real-valued) functions

$$f_1, \ldots, f_T : \mathcal{X} \to \mathbb{R} \tag{1}$$

from corresponding training sets $(x_{it}, y_{it})_{i=1}^{n_t}$ with $x_{it} \in \mathcal{X}$ and $y_{it} \in \mathbb{R}$, for $t = 1, \ldots, T$. The key idea in MTL is to estimate $f_1, \ldots, f_T$ jointly, rather than independently. The intuition is that *if* the different tasks are *related* this strategy can lead to a substantial decrease of sample complexity, that is the amount of data needed to achieve a given accuracy. The crucial question is then how to encode and exploit such relations among the tasks.

Previous work on MTL has mostly focused on studying the case where the tasks are linearly related (see Sec. 5). Indeed, this allows to capture a wide range of relevant situations and the resulting problem can be often cast as a convex optimization, which can be solved efficiently. However, it is not hard to imagine situations where different tasks might be nonlinearly related. As a simple example consider the problem of learning two functions $f_1, f_2 : [0, 2\pi] \to \mathbb{R}$, with $f_1(x) = \cos(x)$ and $f_2(x) = \sin(x)$. Clearly the two tasks are strongly related one to the other (they need to satisfy $f_1(x)^2 + f_2(x)^2 - 1 = 0$ for all $x \in [0, 2\pi]$) but such structure in nonlinear (here an equation of degree 2). More realistic examples can be found for instance in the context of modeling physical systems, such as the case of a robot manipulator. A prototypical learning problem (see e.g. [22]) is to associate the current state of the system (position, velocity, acceleration) to a variety of measurements (e.g. torques) that are nonlinearly related one to the other by physical constraints (see e.g. [23]).

Following the intuition above, in this work we model tasks relations as a set of $P$ equations. Specifically we consider a *constraint function* $\gamma : \mathbb{R}^T \to \mathbb{R}^P$ and require that $\gamma(f_1(x), \ldots, f_T(x)) = 0$ for all $x \in \mathcal{X}$. When $\gamma$ is linear, the problem reverts to linear MTL and can be addressed via standard approaches (see Sec. 5). On the contrary, the nonlinear case becomes significantly more challenging and it is not clear how to address it in general. The starting point of our study is to consider the tasks predictors as a vector-valued function $f = (f_1, \ldots, f_T) : \mathcal{X} \to \mathbb{R}^T$ but then observe that $\gamma$ imposes constraints on its range. Specifically, in this work we restrict $f : \mathcal{X} \to \mathcal{C}$ to take values in the *constraint set*

$$\mathcal{C} = \left\{ y \in \mathbb{R}^T \mid \gamma(y) = 0 \right\} \subseteq \mathbb{R}^T \tag{2}$$

and formulate the nonlinear multitask learning problem as that of finding a good approximation $\widehat{f} : \mathcal{X} \to \mathcal{C}$ to the solution of the multi-task *expected risk* minimization problem

$$\underset{f:\mathcal{X}\to\mathcal{C}}{\text{minimize}} \; \mathcal{E}(f), \qquad \mathcal{E}(f) = \frac{1}{T} \sum_{t=1}^{T} \int_{\mathcal{X}\times\mathbb{R}} \ell(f_t(x), y) d\rho_t(x, y) \tag{3}$$

where $\ell : \mathbb{R} \times \mathbb{R} \to \mathbb{R}$ is a prescribed loss function measuring prediction errors for each individual task and, for every $t = 1, \dots, T$, $\rho_t$ is the distribution on $\mathcal{X} \times \mathbb{R}$ from which the training points $(x_{it}, y_{it})_{i=1}^{n_t}$ have been independently sampled.

Nonlinear MTL poses several challenges to standard machine learning approaches. Indeed, when $\mathcal{C}$ is a linear space (e.g. $\gamma$ is a linear map) the typical strategy to tackle problem (3) is to minimize the *empirical risk* $\frac{1}{T} \sum_{t=1}^{T} \frac{1}{n_t} \sum_{i=1}^{n_t} \ell(f_t(x_{it}), y_{it})$ over some suitable space of hypotheses $f : \mathcal{X} \to \mathcal{C}$ within which optimization can be performed efficiently. However, if $\mathcal{C}$ is a nonlinear subset of $\mathbb{R}^T$, it is not clear how to parametrize a "good" space of functions since most basic properties typically needed by optimization algorithms are lost (e.g. $f_1, f_2 : \mathcal{X} \to \mathcal{C}$ does not necessarily imply $f_1 + f_2 : \mathcal{X} \to \mathcal{C}$). To address this issue, in this paper we adopt the *structured prediction* perspective proposed in [21], which we review in the following.

## 2.1   Background: Structured Prediction and the SELF Framework

The term structured prediction typically refers to supervised learning problems with discrete outputs, such as strings or graphs [19, 20, 24]. The framework in [21] generalizes this perspective to account for a more flexible formulation of structured prediction where the goal is to learn an estimator approximating the minimizer of

$$\underset{f:\mathcal{X} \to \mathcal{C}}{\text{minimize}} \int_{\mathcal{X} \times \mathcal{Y}} \mathcal{L}(f(x), y) d\rho(x, y) \tag{4}$$

given a training set $(x_i, y_i)_{i=1}^{n}$ of points independently sampled from an unknown distribution $\rho$ on $\mathcal{X} \times \mathcal{Y}$, where $\mathcal{L} : \mathcal{Y} \times \mathcal{Y} \to \mathbb{R}$ is a loss function. The output sets $\mathcal{Y}$ and $\mathcal{C} \subseteq \mathcal{Y}$ are not assumed to be linear spaces but can be either discrete (e.g. strings, graphs, etc.) or dense (e.g. manifolds, distributions, etc.) sets of "structured" objects. This generalization will be key to tackle the question of multitask learning with nonlinear output relations in Sec. 3 since it allows to consider the case where $\mathcal{C}$ is a generic subset of $\mathcal{Y} = \mathbb{R}^T$. The analysis in [21] hinges on the assumption that the loss $\mathcal{L}$ is "bi-linearizable", namely

**Definition 1** (SELF). *Let $\mathcal{Y}$ be a compact set. A function $\ell : \mathcal{Y} \times \mathcal{Y} \to \mathbb{R}$ is a* Structure Encoding Loss Function (SELF) *if there exists a continuous feature map $\psi : \mathcal{Y} \to \mathcal{H}$, with $\mathcal{H}$ a reproducing kernel Hilbert space on $\mathcal{Y}$ and a continuous linear operator $V : \mathcal{H} \to \mathcal{H}$ such that for all $y, y' \in \mathcal{Y}$*

$$\ell(y, y') = \langle \psi(y), V\psi(y') \rangle_{\mathcal{H}}. \tag{5}$$

In the original work the SELF definition was dubbed "loss trick" as a parallel to the *kernel trick* [25]. As we discuss in Sec. 4, most MTL loss functions indeed satisfy the SELF property. Under this assumption, it can be shown that a solution $f^* : \mathcal{X} \to \mathcal{C}$ to Eq. (4) must satisfy

$$f^*(x) = \underset{c \in \mathcal{C}}{\operatorname{argmin}} \ \langle \psi(c), V \ g^*(x) \rangle_{\mathcal{H}} \qquad \text{with} \qquad g^*(x) = \int_{\mathcal{Y}} \psi(y) \ d\rho(y|x) \tag{6}$$

for all $x \in \mathcal{X}$ (see [21] or the Appendix). Since $g^* : \mathcal{X} \to \mathcal{H}$ is a function with values in a linear space, we can apply standard regression techniques to learn a $\widehat{g} : \mathcal{X} \to \mathcal{H}$ to approximate $g^*$ given $(x_i, \psi(y_i))_{i=1}^{n}$ and then obtain the estimator $\widehat{f} : \mathcal{X} \to \mathcal{C}$ as

$$\widehat{f}(x) = \underset{c \in \mathcal{C}}{\operatorname{argmin}} \ \langle \psi(c) \,, V \, \widehat{g}(x) \rangle_{\mathcal{H}} \qquad \forall x \in \mathcal{X}. \tag{7}$$

The intuition here is that if $\widehat{g}$ is close to $g^*$, so it will be $\widehat{f}$ to $f^*$ (see Sec. 4 for a rigorous analysis of this relation). If $\widehat{g}$ is the *kernel ridge regression* estimator obtained by minimizing the empirical risk $\frac{1}{n} \sum_{i=1}^{n} \|g(x_i) - \psi(y_i)\|_{\mathcal{H}}^2$ (plus regularization), Eq. (7) becomes

$$\widehat{f}(x) = \underset{c \in \mathcal{C}}{\operatorname{argmin}} \ \sum_{i=1}^{n} \alpha_i(x) \mathcal{L}(c, y_i), \qquad \alpha(x) = (\alpha_1(x), \dots, \alpha_n(x))^\top = (K + n\lambda I)^{-1} K_x \tag{8}$$

since $\widehat{g}$ can be written as the linear combination $\widehat{g}(x) = \sum_{i=1}^{n} \alpha_i(x) \ \psi(y_i)$ and the loss function $\mathcal{L}$ is SELF. In the above formula $\lambda > 0$ is a hyperparameter, $I \in \mathbb{R}^{n \times n}$ the identity matrix, $K \in \mathbb{R}^{n \times n}$ the kernel matrix with elements $K_{ij} = k(x_i, x_j)$, $K_x \in \mathbb{R}^n$ the vector with entries $(K_x)_i = k(x, x_i)$ and $k : \mathcal{X} \times \mathcal{X} \to \mathbb{R}$ a reproducing kernel on $\mathcal{X}$.

The SELF structured prediction approach is therefore conceptually divided into two distinct phases: a *learning* step, where the score functions $\alpha_i : \mathcal{X} \to \mathbb{R}$ are estimated, which consists in solving the kernel ridge regression in $\widehat{g}$, followed by a *prediction* step, where the vector $c \in \mathcal{C}$ minimizing the weighted sum in Eq. (8) is identified. Interestingly, while the feature map $\psi$, the space $\mathcal{H}$ and the operator $V$ allow to derive the SELF estimator, *their knowledge is not needed to evaluate $\widehat{f}(x)$ in practice* since the optimization at Eq. (8) depends exclusively on the loss $\mathcal{L}$ and the score functions $\alpha_i$.

## 3 Structured Prediction for Nonlinear MTL

In this section we present the main contribution of this work, namely the extension of the SELF framework to the MTL setting. Furthermore, we discuss how to cope with possible violations of the constraint set in practice. We study the theoretical properties of the proposed estimator in Sec. 4. We begin our analysis by applying the SELF approach to vector-valued regression which will then lead to the MTL formulation.

### 3.1 Nonlinear Vector-valued Regression

Vector-valued regression (VVR) is a special instance of MTL where for each input, all output examples are available during training. In other words, the training sets can be combined into a single dataset $(x_i, y_i)_{i=1}^n$, with $y_i = (y_{i1}, \ldots, y_{it})^\top \in \mathbb{R}^T$. If we denote $\mathcal{L} : \mathbb{R}^T \times \mathbb{R}^T \to \mathbb{R}$ the separable loss $\mathcal{L}(y, y') = \frac{1}{T}\sum_{t=1}^T \ell(y_t, y'_t)$, nonlinear VVR coincides with the structured prediction problem in Eq. (4). If $\mathcal{L}$ is SELF, we can therefore obtain an estimator according to Eq. (8).

**Example 1** (Nonlinear VVR with Square Loss). *Let $\mathcal{L}(y, y') = \sum_{t=1}^T (y_t - y'_t)^2$. Then, the SELF estimator for nonlinear VVR can be obtained as $\widehat{f} : \mathcal{X} \to \mathcal{C}$ from Eq. (8) and corresponds to the projection onto $\mathcal{C}$*

$$\widehat{f}(x) = \operatorname*{argmin}_{c \in \mathcal{C}} \ \|c - b(x)/a(x)\|_2^2 = \Pi_{\mathcal{C}}\left(b(x)/a(x)\right) \tag{9}$$

*with $a(x) = \sum_{i=1}^n \alpha_i(x)$ and $b(x) = \sum_{i=1}^n \alpha_i(x)\, y_i$. Interestingly, $b(x) = \sum_{i=1}^n \alpha_i(x)y_i = Y^\top(K + n\lambda I)^{-1}K_x$ corresponds to the solution of the standard vector-valued kernel ridge regression without constraints (we denoted $Y \in \mathbb{R}^{n \times T}$ the matrix with rows $y_i^\top$). Therefore, nonlinear VVR consists in:* 1) *computing the* unconstrained *kernel ridge regression estimator $b(x)$,* 2) *normalizing it by $a(x)$ and* 3) *projecting it onto $\mathcal{C}$.*

The example above shows that for specific loss functions the estimation of $\widehat{f}(x)$ can be significantly simplified. In general, such optimization will depend on the properties of the constraint set $\mathcal{C}$ (e.g. convex, connected, etc.) and the loss $\ell$ (e.g. convex, smooth, etc.). In practice, if $\mathcal{C}$ is a discrete (or discretized) subset of $\mathbb{R}^T$, the computation can be performed efficiently via a nearest neighbor search (e.g. using *k-d trees* based approaches to speed up computations [26]). If $\mathcal{C}$ is a manifold, recent *geometric optimization* methods [27] (e.g. SVRG [28]) can be applied to find critical points of Eq. (8). This setting suggests a connection with manifold regression as discussed below.

**Remark 1** (Connection to Manifold Regression). *When $\mathcal{C}$ is a Riemannian manifold, the problem of learning $f : \mathcal{X} \to \mathcal{C}$ shares some similarities to the* manifold regression *setting studied in [17] (see also [29] and references therein). Manifold regression can be interpreted as a vector-valued learning setting where outputs are constrained to be in $\mathcal{C} \subseteq \mathbb{R}^T$ and prediction errors are measured according to the* geodesic distance. *However, note that the two problems are also significantly different since,* 1) *in MTL noise could make output examples $y_i$ lie close but not exactly on the constraint set $\mathcal{C}$ and moreover,* 2) *the loss functions used in MTL typically measure errors independently for each task (as in Eq. (3), see also [5]) and rarely coincide with a geodesic distance.*

### 3.2 Nonlinear Multitask Learning

Differently from nonlinear vector-valued regression, the SELF approach introduced in Sec. 2.1 *cannot* be applied to the MTL setting. Indeed, the estimator at Eq. (8) requires knowledge of all tasks outputs $y_i \in \mathcal{Y} = \mathbb{R}^T$ for every training input $x_i \in \mathcal{X}$ while in MTL we have a separate dataset $(x_{it}, y_{it})_{i=1}^{n_t}$ for each task, with $y_{it} \in \mathbb{R}$ (this could be interpreted as the vector $y_i$ to have "missing entries").

Therefore, in this work we extend the SELF framework to nonlinear MTL. We begin by proving a characterization of the minimizer $f^* : \mathcal{X} \to \mathcal{C}$ of the expected risk $\mathcal{E}(f)$ akin to Eq. (6).

**Proposition 2.** *Let $\ell : \mathbb{R} \times \mathbb{R} \to \mathbb{R}$ be SELF, with $\ell(y, y') = \langle \psi(y), V\psi(y') \rangle_{\mathcal{H}}$. Then, the expected risk $\mathcal{E}(f)$ introduced at Eq. (3) admits a measurable minimizer $f^* : \mathcal{X} \to \mathcal{C}$. Moreover, any such minimizer satisfies, almost everywhere on $\mathcal{X}$,*

$$f^*(x) = \operatorname*{argmin}_{c \in \mathcal{C}} \sum_{t=1}^{T} \langle \psi(c_t), Vg_t^*(x) \rangle_{\mathcal{H}}, \qquad with \qquad g_t^*(x) = \int_{\mathbb{R}} \psi(y)\, d\rho_t(y|x). \tag{10}$$

Prop. 2 extends Eq. (6) by relying on the linearity induced by the SELF assumption combined with the *Aumann's principle* [30], which guarantees the existence of a measurable selector $f^*$ for the minimization problem at Eq. (10) (see Appendix). By following the strategy outlined in Sec. 2.1, we propose to learn $T$ independent functions $\widehat{g}_t : \mathcal{X} \to \mathcal{H}$, each aiming to approximate the corresponding $g_t^* : \mathcal{X} \to \mathcal{H}$ and then define $\widehat{f} : \mathcal{X} \to \mathcal{C}$ such that

$$\widehat{f}(x) = \operatorname*{argmin}_{c \in \mathcal{C}} \sum_{t=1}^{T} \langle\, \psi(c_t)\,,\ V\,\widehat{g}_t(x)\, \rangle_{\mathcal{H}} \qquad \forall x \in \mathcal{X}. \tag{11}$$

We choose the $\widehat{g}_t$ to be the solutions to $T$ independent kernel ridge regressions problems

$$\underset{g \in \mathcal{H} \otimes \mathcal{G}}{\text{minimize}}\ \frac{1}{n_t} \sum_{i=1}^{n_t} \|g(x_{it}) - \psi(y_{it})\|^2 + \lambda_t \|g\|_{\mathcal{H} \otimes \mathcal{G}}^2 \tag{12}$$

for $t = 1, \ldots, T$, where $\mathcal{G}$ is a reproducing kernel Hilbert space on $\mathcal{X}$ associated to a kernel $k : \mathcal{X} \times \mathcal{X} \to \mathbb{R}$ and the candidate solution $g : \mathcal{X} \to \mathcal{H}$ is an element of $\mathcal{H} \otimes \mathcal{G}$. The following result shows that in this setting, evaluating the estimator $\widehat{f}$ can be significantly simplified.

**Proposition 3** (The Nonlinear MTL Estimator)**.** *Let $k : \mathcal{X} \times \mathcal{X} \to \mathbb{R}$ be a reproducing kernel with associated reproducing kernel Hilbert space $\mathcal{G}$. Let $\widehat{g}_t : \mathcal{X} \to \mathcal{H}$ be the solution of Eq. (12) for $t = 1, \ldots, T$. Then the estimator $\widehat{f} : \mathcal{X} \to \mathcal{C}$ defined at Eq. (11) is such that*

$$\widehat{f}(x) = \operatorname*{argmin}_{c \in \mathcal{C}} \sum_{t=1}^{T} \sum_{i=1}^{n_t} \alpha_{it}(x)\ell(c_t, y_{it}), \quad (\alpha_{1t}(x), \ldots, \alpha_{n_t t}(x))^{\top} = (K_t + n_t \lambda_t I)^{-1} K_{tx} \tag{13}$$

*for all $x \in \mathcal{X}$ and $t = 1, \ldots, T$, where $K_t \in \mathbb{R}^{n_t \times n_t}$ denotes the kernel matrix of the $t$-th task, namely $(K_t)_{ij} = k(x_{it}, x_{jt})$, and $K_{tx} \in \mathbb{R}^{n_t}$ the vector with $i$-th component equal to $k(x, x_{it})$.*

Prop. 3 provides an equivalent characterization for nonlinear MTL estimator at Eq. (11) that is more amenable to computations (it does not require explicit knowledge of $\mathcal{H}$, $\psi$ or $V$) and generalizes the SELF approach (indeed for VVR, Eq. (13) reduces to the SELF estimator at Eq. (8)). Interestingly, the proposed strategy learns the score functions $\alpha_{im} : \mathcal{X} \to \mathbb{R}$ separately for each task and then combines them in the joint minimization over $\mathcal{C}$. This can be interpreted as the estimator weighting predictions according to how "reliable" each task is on the input $x \in \mathcal{X}$. We make this intuition more clear in the following.

**Example 2** (Nonlinear MTL with Square Loss)**.** *Let $\ell$ be the square loss. Then, analogously to Example 1 we have that for any $x \in \mathcal{X}$, the multitask estimator at Eq. (13) is*

$$\widehat{f}(x) = \operatorname*{argmin}_{c \in \mathcal{C}} \sum_{t=1}^{T} a_t(x)\big(c_t - b_t(x)/a_t(x)\big)^2 \tag{14}$$

*with $a_t(x) = \sum_{i=1}^{n_t} \alpha_{it}(x)$ and $b_t(x) = \sum_{i=1}^{n_t} \alpha_{it}(x)y_{it}$, which corresponds to perform the projection $\widehat{f}(x) = \Pi_{\mathcal{C}}^{A(x)}(w(x))$ of the vector $w(x) = (b_1(x)/a_1(x), \ldots, b_T(x)/a_T(x))$ according to the metric deformation induced by the matrix $A(x) = diag(a_1(x), \ldots, a_T(x))$. This suggests to interpret $a_t(x)$ as a measure of confidence of task $t$ with respect to $x \in \mathcal{X}$. Indeed, tasks with small $a_t(x)$ will affect less the weighted projection $\Pi_{\mathcal{C}}^{A(x)}$.*

### 3.3 Extensions: Violating $\mathcal{C}$

In practice, it is natural to expect the knowledge of the constraints set $\mathcal{C}$ to be not exact, for instance due to noise or modeling inaccuracies. To address this issue, we consider two extensions of nonlinear MTL that allow candidate predictors to slightly violate the constraints $\mathcal{C}$ and introduce a hyperparameter to control this effect.

**Robustness w.r.t. perturbations of $\mathcal{C}$.** We soften the effect of the constraint set by requiring candidate predictors to take value *within* a radius $\delta > 0$ from $\mathcal{C}$, namely $f : \mathcal{X} \to \mathcal{C}_\delta$ with

$$\mathcal{C}_\delta = \{ c + r \mid c \in \mathcal{C}, r \in \mathbb{R}^T, \|r\| \le \delta \}. \tag{15}$$

The scalar $\delta > 0$ is now a hyperparameter ranging from $0$ ($\mathcal{C}_0 = \mathcal{C}$) to $+\infty$ ($\mathcal{C}_\infty = \mathbb{R}^T$).

**Penalizing w.r.t. the distance from $\mathcal{C}$.** We can penalize predictions depending on their distance from the set $\mathcal{C}$ by introducing a perturbed version $\ell_\mu^t : \mathbb{R}^T \times \mathbb{R}^T \to \mathbb{R}$ of the loss

$$\ell_\mu^t(y, z) = \ell(y_t, z_t) + \|z - \Pi_C(z)\|^2 / \mu \qquad \text{for all } y, z \in \mathbb{R}^T \tag{16}$$

where $\Pi_\mathcal{C} : \mathbb{R}^T \to \mathcal{C}$ denotes the orthogonal projection onto $\mathcal{C}$ (see Example 1). Below we report the closed-from solution for nonlinear vector-valued regression with square loss.

**Example 3** (VVR and Violations of $\mathcal{C}$). *With the same notation as Example 1, let $f_0 : \mathcal{X} \to \mathcal{C}$ denote the solution at Eq. (9) of nonlinear VVR with* exact *constraints, let $r = b(x)/a(x) - f_0(x) \in \mathbb{R}^T$. Then, the solutions to the problem with robust constraints $\mathcal{C}_\delta$ and perturbed loss function $\mathcal{L}_\mu = \frac{1}{T} \sum_t \ell_\mu^t$ are respectively (see Appendix for the MTL)*

$$\widehat{f}_\delta(x) = f_0(x) + r \, \min(1, \delta / \|r\|) \qquad and \qquad \widehat{f}_\mu(x) = f_0(x) + r \, \mu / (1 + \mu). \tag{17}$$

## 4 Generalization Properties of Nonlinear MTL

We now study the statistical properties of the proposed nonlinear MTL estimator. Interestingly, this will allow to identify specific training regimes in which nonlinear MTL achieves learning rates significantly faster than those available when learning the tasks independently. Our analysis revolves around the assumption that the loss function used to measure prediction errors is SELF. To this end we observe that most multitask loss functions are indeed SELF.

**Proposition 4.** *Let $\bar{\ell} : [a, b] \to \mathbb{R}$ be differentiable almost everywhere with derivative Lipschitz continuous almost everywhere. Let $\ell : [a, b] \times [a, b] \to \mathbb{R}$ be such that $\ell(y, y') = \bar{\ell}(y - y')$ or $\ell(y, y') = \bar{\ell}(yy')$ for all $y, y' \in \mathbb{R}$. Then: $(i)$ $\ell$ is SELF and $(ii)$ the separable function $\mathcal{L} : \mathcal{Y}^T \times \mathcal{Y}^T \to \mathbb{R}$ such that $\mathcal{L}(y, y') = \frac{1}{T} \sum_{t=1}^{T} \ell(y_t, y_t')$ for all $y, y' \in \mathcal{Y}^T$ is SELF.*

Interestingly, most (mono-variate) loss functions used in multitask and supervised learning satisfy the assumptions of Prop. 4. Typical examples are the square loss $(y - y')^2$, hinge $\max(0, 1 - yy')$ or logistic $\log(1 - \exp(-yy'))$: the corresponding derivative with respect to $z = y - y'$ or $z = yy'$ exists and it is Lipschitz almost everywhere on compact sets.

The nonlinear MTL estimator introduced in Sec. 3.2 relies on the intuition that if for each $x \in \mathcal{X}$ the kernel ridge regression solutions $\widehat{g}_t(x)$ are close to the conditional expectations $g_t^*(x)$, then also $\widehat{f}(x)$ will be close to $f^*(x)$. The following result formally characterizes the relation between the two problems, proving what is often referred to as a *comparison inequality* in the context of surrogate frameworks [31]. Throughout the rest of this section we assume $\rho_t(x, y) = \rho_t(y|x)\rho_\mathcal{X}(x)$ for each $t = 1, \ldots, T$ and denote $\|g\|_{L_{\rho_\mathcal{X}}^2}$ the $L_{\rho_\mathcal{X}}^2 = L^2(\mathcal{X}, \mathcal{H}, \rho_\mathcal{X})$ norm of a function $g : \mathcal{X} \to \mathcal{H}$ according to the marginal distribution $\rho_\mathcal{X}$.

**Theorem 5** (Comparison Inequality). *With the same assumptions of Prop. 2, for $t = 1, \ldots, T$ let $f^* : \mathcal{X} \to \mathcal{C}$ and $g_t^* : \mathcal{X} \to \mathcal{H}$ be defined as in Eq. (10), let $\widehat{g}_t : \mathcal{X} \to \mathcal{H}$ be measurable functions and let $\widehat{f} : \mathcal{X} \to \mathcal{C}$ satisfy Eq. (11). Let $V^*$ be the adjoint of $V$. Then,*

$$\mathcal{E}(\widehat{f}) - \mathcal{E}(f^*) \le q_{\mathcal{C}, \ell, T} \sqrt{\frac{1}{T} \sum_{t=1}^{T} \|\widehat{g}_t - g_t^*\|_{L_{\rho_\mathcal{X}}^2}^2}, \quad q_{\mathcal{C}, \ell, T} = 2 \sup_{c \in \mathcal{C}} \sqrt{\frac{1}{T} \sum_{t=1}^{T} \|V^* \psi(c_t)\|_\mathcal{H}^2}. \tag{18}$$

The comparison inequality at Eq. (18) is key to study the generalization properties of our nonlinear MTL estimator by showing that we can control its *excess risk* in terms of how well the $\widehat{g}_t$ approximate the true $g_t^*$ (see Appendix for a proof of Thm. 5).

**Theorem 6.** *Let $\mathcal{C} \subseteq [a, b]^T$, let $\mathcal{X}$ be a compact set and $k : \mathcal{X} \times \mathcal{X} \to \mathbb{R}$ a continuous universal reproducing kernel (e.g. Gaussian). Let $\ell : [a, b] \times [a, b] \to \mathbb{R}$ be a SELF. Let $\widehat{f}_N : \mathcal{X} \to \mathcal{C}$ denote the estimator at Eq. (13) with $N = (n_1, \ldots, n_T)$ training points independently sampled from $\rho_t$ for each task $t = 1, \ldots, T$ and $\lambda_t = n_t^{-1/4}$. Let $n_0 = \min_{1 \leq t \leq T} n_t$. Then, with probability 1*

$$\lim_{n_0 \to +\infty} \mathcal{E}(\widehat{f}_N) = \inf_{f : \mathcal{X} \to \mathcal{C}} \mathcal{E}(f). \tag{19}$$

The proof of Thm. 6 relies on the comparison inequality in Thm. 5, which links the excess risk of the MTL estimator to the square error between $\hat{g}_t$ and $g_t^*$. Standard results from kernel ridge regression allow to conclude the proof [32] (see a more detailed discussion in the Appendix). By imposing further standard assumptions, we can also obtain generalization bounds on $\|\widehat{g}_t - g_t^*\|_{L^2}$ that automatically apply to nonlinear MTL again via the comparison inequality, as shown below.

**Theorem 7.** *With the same assumptions and notation of Thm. 6 let $\widehat{f}_N : \mathcal{X} \to \mathcal{C}$ denote the estimator at Eq. (13) with $\lambda_t = n_t^{-1/2}$ and assume $g_t^* \in \mathcal{H} \otimes \mathcal{G}$, for all $t = 1, \ldots, T$. Then for any $\tau > 0$ we have, with probability at least $1 - 8e^{-\tau}$, that*

$$\mathcal{E}(\widehat{f}_N) - \inf_{f : \mathcal{X} \to \mathcal{C}} \mathcal{E}(f) \leq q_{\mathcal{C}, \ell, T} \ h_\ell \ \tau^2 \ n_0^{-1/4} \log T, \tag{20}$$

*where $q_{\mathcal{C}, \ell, T}$ is defined as in Eq. (18) and $h_\ell$ is a constant independent of $\mathcal{C}, N, n_t, \lambda_t, \tau, T$.*

The the excess risk bound in Thm. 7 is comparable to that in [21] (Thm. 5). To our knowledge this is the first result studying the generalization properties of a learning approach to MTL with constraints.

## 4.1 Benefits of Nonlinear MTL

The rates in Thm. 7 strongly depend on the constraints $\mathcal{C}$ via the constant $q_{\mathcal{C}, \ell, T}$. The following result studies two special cases that allow to appreciate this effect.

**Lemma 8.** *Let $B \geq 1$, $\mathcal{B} = [-B, B]^T$, $\mathcal{S} \subset \mathbb{R}^T$ be the sphere of radius $B$ centered at the origin and let $\ell$ be the square loss. Then $q_{\mathcal{B}, \ell, T} \leq 2\sqrt{5} \ B^2$ and $q_{\mathcal{S}, \ell, T} \leq 2\sqrt{5} \ B^2 \ T^{-1/2}$.*

To explain the effect of $\mathcal{C}$ on MTL, define $n = \sum_{t=1}^{T} n_t$ and assume that $n_0 = n_t = n/T$. Lemma 8 together with Thm. 7 shows that when the tasks are assumed not to be related (i.e. $\mathcal{C} = \mathcal{B}$) the learning rate of nonlinear MTL is of $\widetilde{O}((\frac{T}{n})^{1/4})$, as if the tasks were learned independently. On the other hand, when the tasks have a relation (e.g. $\mathcal{C} = \mathcal{S}$, implying a quadratic relation between the tasks) nonlinear MTL achieves a learning rate of $\widetilde{O}((\frac{1}{nT})^{1/4})$, which improves as the number of tasks increases and as the total number of observed examples increases. Specifically, for $T$ of the same order of $n$, we obtain a rate of $\widetilde{O}(n^{-1/2})$ which is comparable to the optimal rates available for kernel ridge regression *with only one task trained on the total number $n$ of examples* [32]. This observation corresponds to the intuition that if we have many related tasks with few training examples each, we can expect to achieve significantly better generalization by taking advantage of such relations rather than learning each task independently.

## 5 Connection to Previous Work: Linear MTL

In this work we formulated the nonlinear MTL problem as that of learning a function $f : \mathcal{X} \to \mathcal{C}$ taking values in a set of constraints $\mathcal{C} \subseteq \mathbb{R}^T$ *implicitly* identified by a set of equations $\gamma(f(x)) = 0$. An alternative approach would be to characterize the set $\mathcal{C}$ via an *explicit* parametrization $\theta : \mathbb{R}^Q \to \mathcal{C}$, for $Q \in \mathbb{N}$, so that the multitask predictor can be decomposed as $f = \theta \circ h$, with $h : \mathcal{X} \to \mathbb{R}^Q$. We can learn $h : \mathcal{X} \to \mathbb{R}^Q$ using empirical risk minimization strategies such as Tikhonov regularization,

$$\underset{h=(h_1, \ldots, h_Q) \in \mathcal{H}^Q}{\text{minimize}} \quad \frac{1}{n} \sum_{i=1}^{n} \mathcal{L}(\theta \circ h(x_i), y_i) + \lambda \sum_{q=1}^{Q} \|h_q\|_{\mathcal{H}}^2 \tag{21}$$

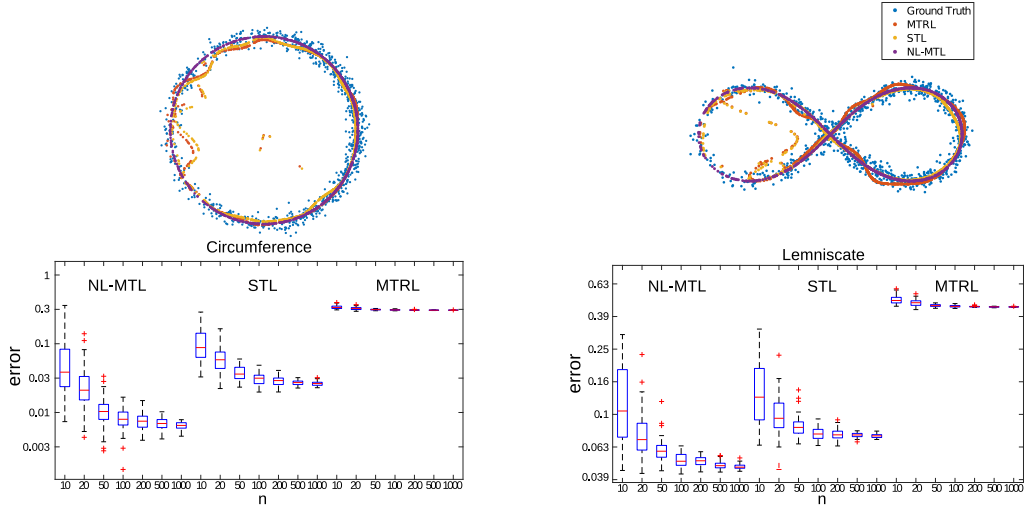

Figure 1: (**Bottom**) MSE (logaritmic scale) of MTL methods for learning constrained on a circumference (Left) or a Lemniscate (Right). Results are reported in a boxplot across 10 trials. (**Top**) Sample predictions of the three methods trained on 100 points and compared with the ground truth.

since candidate $h$ take value in $\mathbb{R}^Q$ and therefore $\mathcal{H}$ can be a standard linear space of hypotheses. However, while Eq. (21) is interesting from the modeling standpoint, it also poses several problems: 1) $\theta$ can be nonlinear or even non-continuous, making Eq. (21) hard to solve in practice even for $\mathcal{L}$ convex; 2) $\theta$ is not uniquely identified by $\mathcal{C}$ and therefore different parametrizations may lead to very different $\widehat{f} = \theta \circ \widehat{h}$, which is not always desirable; 3) There are few results on empirical risk minimization applied to generic loss functions $\mathcal{L}(\theta(\cdot), \cdot)$ (via so-called oracle inequalities, see [30] and references therein), and it is unclear what generalization properties to expect in this setting. A relevant exception to the issues above is the case where $\theta$ is *linear*. In this setting Eq. (21) becomes more amenable to both computations and statistical analysis and indeed most previous MTL literature has been focused on this setting, either by designing ad-hoc output metrics [33], linear output encodings [34] or regularizers [5]. Specifically, in this latter case the problem is cast as that of minimizing the functional

$$\underset{f=(f_1,\ldots,f_T)\in\mathcal{H}^T}{\text{minimize}} \quad \sum_{i=1}^{n} \mathcal{L}(f(x_i), y_i) + \lambda \sum_{t,s=1}^{T} A_{ts}\langle f_t, f_s\rangle_{\mathcal{H}} \tag{22}$$

where the psd matrix $A = (A_{ts})_{s,t=1}^{T}$ encourages linear relations between the tasks. It can be shown that this problem is equivalent to Eq. (21) when the $\theta \in \mathbb{R}^{T \times Q}$ is linear and $A$ is set to the pseudoinverse of $\theta\theta^\top$. As shown in [14], a variety of situations are recovered considering the approach above, such as the case where tasks are centered around a common average [9], clustered in groups [10] or sharing the same subset of features [3, 35]. Interestingly, the above framework can be further extended to estimate the structure matrix $A$ directly from data, an idea initially proposed in [12] and further developed in [2, 14, 16].

## 6 Experiments

**Synthetic Dataset**. We considered a model of the form $y = f^*(x) + \epsilon$, with $\epsilon \sim \mathbb{N}(0, \sigma I)$ noise sampled according to a normal distribution and $f^* : \mathcal{X} \to \mathcal{C}$, where $\mathcal{C} \subset \mathbb{R}^2$ was either a circumference or a lemniscate (see Fig. 1) of equation $\gamma_{\text{circ}}(y) = y_1^2 + y_2^2 - 1 = 0$ and $\gamma_{\text{lemn}}(y) = y_1^4 - (y_1^2 - y_2^2) = 0$ for $y \in \mathbb{R}^2$. We set $\mathcal{X} = [-\pi, \pi]$ and $f^*_{\text{circ}}(x) = (\cos(x), \sin(x))$ or $f^*_{\text{lemn}}(x) = (\sin(x), \sin(2x))$ the parametric functions associated respectively to the circumference and Lemniscate. We sampled from 10 to 1000 points for training and 1000 for testing, with noise $\sigma = 0.05$.

We trained and tested three regression models over 10 trials. We used a Gaussian kernel on the input and chose the corresponding bandwidth and the regularization parameter $\lambda$ by hold-out cross-validation on 30% of the training set (see details in the appendix). Fig. 1 (Bottom) reports the mean

Table 1: Explained variance of the robust (NL-MTL[R]) and perturbed (NL-MTL[P]) variants of nonlinear MTL, compared with linear MTL methods on the Sarcos dataset reported from [16].

| | STL | MTL[36] | CMTL[10] | MTRL[11] | MTFL[13] | FMTL[16] | NL-MTL[R] | NL-MTL[P] |
|---|---|---|---|---|---|---|---|---|
| **Expl. Var. (%)** | 40.5 ±7.6 | 34.5 ±10.2 | 33.0 ±13.4 | 41.6 ±7.1 | 49.9 ±6.3 | 50.3 ±5.8 | **55.4 ±6.5** | 54.6 ±5.1 |

Table 2: Rank prediction error according to the weighted binary loss in [37, 21].

| | NL-MTL | SELF[21] | Linear [37] | Hinge [38] | Logistic [39] | SVMStruct [20] | STL | MTRL[11] |
|---|---|---|---|---|---|---|---|---|
| **Rank Loss** | **0.271 ±0.004** | 0.396 ±0.003 | 0.430 ±0.004 | 0.432 ±0.008 | 0.432 ±0.012 | 0.451 ±0.008 | 0.581 0.003 | 0.613 ±0.005 |

square error (MSE) of our nonlinear MTL approach (NL-MTL) compared with the standard least squares single task learning (STL) baseline and the multitask relations learning (MTRL) from [11], which encourages tasks to be *linearly* dependent. However, for both circumference and Lemniscate, the tasks are strongly *nonlinearly* related. As a consequence our approach consistently outperforms its two competitors which assume only linear relations (or none at all). Fig. 1 (Top) provides a qualitative comparison on the three methods (when trained with 100 examples) during a single trial.

**Sarcos Dataset**. We report experiments on the Sarcos dataset [22]. The goal is to predict the torque measured at each joint of a 7 degrees-of-freedom robotic arm, given the current state, velocities and accelerations measured at each joint (7 tasks/torques for 21-dimensional input). We used the 10 dataset splits available online for the dataset in [13], each containing 2000 examples per task with 15 examples used for training/validation while the rest is used to measure errors in terms of the *explained variance*, namely 1 - nMSE (as a percentage). To compare with results in [13] we used the linear kernel on the input. We refer to the Appending for details on model selection.

Tab. 1 reports results from [13, 16] for a wide range of previous *linear* MTL methods [36, 10, 3, 11, 13, 16], together with our NL-MTL approach (both robust and perturbed versions). Since, we did not find Sarcos robot model parameters online, we approximated the constraint set $\mathcal{C}$ as a point cloud by collecting 1000 random output vectors that did not belong to training or test sets in [13] (we sampled them from the original dataset [22]). NL-MTL clearly outperforms the "linear" competitors. Note indeed that the torques measured at different joints of a robot are highly nonlinear (see for instance [23]) and therefore taking such structure into account can be beneficial to the learning process.

**Ranking by Pair-wise Comparison.** We consider a ranking problem formulated withing the MTL setting: given $D$ documents, we learn $T = D(D-1)/2$ functions $f_{p,q} : \mathcal{X} \to \{-1, 0, 1\}$, for each pair of documents $p, q = 1, \ldots, D$ that predict whether one document is more relevant than the other for a given input query $x$. The problem can be formulated as multi-label MTL with 0-1 loss: for a given training query $x$ only some labels $y_{p,q} \in \{-1, 0, 1\}$ are available in output (with 1 corresponding to document $p$ being more relevant than $q$, $-1$ the opposite and 0 that the two are equivalent). We have therefore $T$ separate training sets, one for each task (i.e. pair of documents). Clearly, not all possible combinations of outputs $f : \mathcal{X} \to \{-1, 0, 1\}^T$ are allowed since predictions need to be consistent (e.g. if $p \succ q$ (read "$p$ more relevant than $q$") and $q \succ r$, then we cannot have $r \succ p$). As shown in [37] these constraints are naturally encoded in a set $DAG(D)$ in $\mathbb{R}^T$ of all vectors $G \in \mathbb{R}^T$ that correspond to (the vectorized, upper triangular part of the adjacency matrix of) a Directed Acyclic Graph with $D$ vertices. The problem can be cast in our nonlinear MTL framework with $f : \mathcal{X} \to \mathcal{C} = DAG(D)$ (see Appendix for details on how to perform the projection onto $\mathcal{C}$).

We performed experiments on Movielens100k [40] (movies = documents, users = queries) to compare our NL-MTL estimator with both standard MTL baselines as well as methods designed for ranking problems. We used the (linear) input kernel and the train, validation and test splits adopted in [21] to perform 10 independent trials with 5-fold cross-validation for model selection. Tab. 2 reports the average ranking error and standard deviation of the (weighed) 0-1 loss function considered in [37, 21] for the ranking methods proposed in [38, 39, 37], the SVMStruct estimator [20], the SELF estimator considered in [21] for ranking, the MTRL and STL baseline, corresponding to individual SVMs trained for each pairwise comparison. Results for previous methods are reported from [21]. NL-MTL outperforms all competitors, achieving better performance than the the original SELF estimator. For the sake of brevity we refer to the Appendix for more details on the experiments.

**Acknowledgments**. This work was supported in part by EPSRC grant EP/P009069/1.

## Footnotes

*Work performed while A.R. was at the Istituto Italiano di Tecnologia.

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
