[Supplementary Material · supplementary_material.pdf]

# Supplementary Material:
# Consistent Multitask Learning with Nonlinear Output Relations

Here we collect the proofs of the main results presented in the paper.

## A  Theoretical Analysis

In the following we will assume $\mathcal{X}, \mathcal{C}$ to be Polish spaces, namely separable complete metrizable spaces, equipped with the associated Borel sigma-algebra. In particular we will restrict to the case where $\mathcal{C}$ is a subset of $\mathcal{Y}^T$ with $\mathcal{Y} = [a, b]$ and $a, b \in \mathbb{R}$. For any $y \in \mathcal{Y}^T$ we denote with $y_t$ the $t$-th element of $y$. In the rest of the section, we assume that there exists a probability measure $\rho_{\mathcal{X}} = \rho$ on $\mathcal{X}$ such that $\rho_t(y, x) = \rho_t(y|x)\rho(x)$ for all $t = 1, \ldots, T$.

### A.1  Characterization of the Nonlinear MTL Estimator

In order to prove that a solution of the nonlinear MTL problem has the form described in Prop. 2, namely

$$f^*(x) = \underset{c \in \mathcal{C}}{\operatorname{argmin}} \ \sum_{t=1}^{T} \langle \ \psi(c_t) \, , V \ g_t^*(x) \ \rangle_{\mathcal{H}} \qquad g_t^*(x) = \int_{\mathcal{Y}} \psi(y) \ d\rho_t(y|x)$$

we start by providing an alternative characterization for the expected risk $\mathcal{E}(f)$ in terms of the $g_t^* : \mathcal{X} \to \mathcal{H}$.

**Lemma 9.** *Let $\ell : \mathcal{Y} \times \mathcal{Y} \to \mathbb{R}$ be SELF with $\ell(y, y') = \langle \psi(y), V\psi(y')\rangle_{\mathcal{H}}$ for all $y, y' \in \mathcal{Y}$. Then, for any measurable function $f : \mathcal{X} \to \mathcal{C}$, the expected risk $\mathcal{E}(f)$ defined at Eq. (3) is equal to*

$$\mathcal{E}(f) = \frac{1}{T} \int_{\mathcal{X}} \sum_{t=1}^{T} \langle \psi(f_t(x)), \ V \ g_t^*(x)\rangle_{\mathcal{H}} \ d\rho(x) \tag{23}$$

*Note that by the definition of SELF (Def. 1), $\psi$ is continuous on $\mathcal{Y}$ and therefore $g_t$ is measurable and $\|g_t^*(x)\|_{\mathcal{H}}$ bounded a.e. on $\mathcal{X}$ for all $t = 1, \ldots, T$.*

*Proof.*

$$\mathcal{E}(f) = \frac{1}{T} \sum_{t=1}^{T} \int_{\mathcal{X} \times \mathbb{R}} \ell(f_t(x), y) \ d\rho_t(y, x) \tag{24}$$

$$= \frac{1}{T} \sum_{t=1}^{T} \int_{\mathcal{X} \times \mathbb{R}} \langle \psi_t(f_t(x)), V\psi(y)\rangle_{\mathcal{H}} \ d\rho_t(y, x) \tag{25}$$

$$= \frac{1}{T} \sum_{t=1}^{T} \int_{\mathcal{X}} \int_{\mathbb{R}} \langle \psi(f_t(x)), \ V\psi(y)\rangle_{\mathcal{H}} \ d\rho_t(y|x) \ d\rho(x) \tag{26}$$

$$= \frac{1}{T} \int_{\mathcal{X}} \sum_{t=1}^{T} \left\langle \psi(f_t(x)), \ V \int_{\mathbb{R}} \psi(y) d\rho_t(y|x) \right\rangle_{\mathcal{H}} d\rho(x) \tag{27}$$

$$= \frac{1}{T} \int_{\mathcal{X}} \sum_{t=1}^{T} \langle \psi(f_t(x)), \ V \ g_t^*(x)\rangle_{\mathcal{H}} \ d\rho(x). \tag{28}$$

$\square$

The result above implies that if there exists a function $f^* : \mathcal{X} \to \mathcal{C}$ that minimizes the argument $\sum_{t=1}^{T} \langle \psi(f(x), Vg_t^*\rangle$ in the integral at Eq. (23) almost everywhere on $\mathcal{X}$, then $f^*$ minimizes also the expected risk $\mathcal{E}(f)$. The following result guarantees the existence of such a function.

**Proposition 2.** *Let $\ell : \mathbb{R} \times \mathbb{R} \to \mathbb{R}$ be SELF, with $\ell(y, y') = \langle \psi(y), V\psi(y')\rangle_{\mathcal{H}}$. Then, the expected risk $\mathcal{E}(f)$ introduced at Eq. (3) admits a measurable minimizer $f^* : \mathcal{X} \to \mathcal{C}$. Moreover, any such minimizer satisfies, almost everywhere on $\mathcal{X}$,*

$$f^*(x) = \operatorname*{argmin}_{c \in \mathcal{C}} \sum_{t=1}^{T} \langle \psi(c_t), Vg_t^*(x)\rangle_{\mathcal{H}}, \qquad \text{with} \qquad g_t^*(x) = \int_{\mathbb{R}} \psi(y)\, d\rho_t(y|x). \qquad (10)$$

The result is a direct corollary of the following.

**Lemma 10** ($f^*$ *is a minimizer of $\mathcal{E}$*). *Let $g_1, \ldots, g_T : \mathcal{X} \to \mathcal{H}$ be measurable functions with $\|g_t(x)\|_{\mathcal{H}}$ bounded a.e. and let $\bar{\mathcal{E}}$ be defined as*

$$\bar{\mathcal{E}}(f) = \int_{\mathcal{X}} r(x, f(x))d\rho(x), \quad \text{with} \quad r(x,c) := \frac{1}{T}\sum_{t=1}^{T} \langle \psi(c_t),\, V\, g_t(x)\rangle_{\mathcal{H}_{\mathcal{Y}}}$$

*If $\ell : \mathcal{Y} \times \mathcal{Y} \to \mathbb{R}$ is continuous, then there exists a measurable selector $f^\circ(x) \in \operatorname{argmin}_{c \in \mathcal{C}} r(x,c)$ and a measurable function $m(x) = \min_{c \in \mathcal{C}} r(x,c)$ a.e. such that, we have*

$$\bar{\mathcal{E}}(f^\circ) = \int_X m(x)d\rho(x) = \inf_{f : \mathcal{X} \to \mathcal{C}} \bar{\mathcal{E}}(f),$$

*In particular, by selecting $g_t := g_t^*$ of Eq. (10), we have that $\bar{\mathcal{E}}$ is equal to the expected risk $\mathcal{E}(f)$ in Eq. (3), $f^\circ$ is equal to $f^*$ of Eq. (10) and minimizes $\mathcal{E}$.*

*Proof.* Note that $r(x,c)$ is Charatéodory since $\ell$ is continuous and the $g_t$ are measurable. Then, by the compactness of $\mathcal{C}$, we can invoke *Aumann's Measurable Selection Principle* (see e.g. Lemma A.3.18 in [1]), to guarantee that $m$ is measurable and there exists a measurable $f^\circ : \mathcal{X} \to \mathcal{C}$ such that $r(x, f^\circ(x)) = m(x)$ for any $x \in \mathcal{X}$. Moreover, by definition of $m$, we have $r(x, f^\circ(x)) = m(x) \leq r(x, f(x))$ a.e. on $\mathcal{X}$ for any measurable function $f : \mathcal{X} \to \mathcal{C}$. So we have

$$\bar{\mathcal{E}}(f^\circ) = \int_{\mathcal{X}} r(x, f^\circ(x))d\rho(x) = \int_{\mathcal{X}} m(x)d\rho(x) \leq \int_{\mathcal{X}} r(x, f(x))d\rho(x) = \mathcal{E}(f),$$

for any measurable function $f : \mathcal{X} \to \mathcal{C}$. Then $\mathcal{E}(f) = \inf_{f : \mathcal{X} \to \mathcal{C}} \mathcal{E}(f)$. $\qquad\qquad \square$

**Nonlinear Multitask Learning**. Following the intuition provided in Sec. 2.1 for the original SELF algorithm, a natural approach to approximate $f^* : \mathcal{X} \to \mathcal{C}$ is to consider the estimators $\widehat{g}_t : \mathcal{X} \to \mathcal{H}$ for the individual $g_t^* : \mathcal{X} \to \mathcal{H}$ and then define $\widehat{f} : \mathcal{X} \to \mathcal{C}$ such that

$$\widehat{f}(x) = \operatorname*{argmin}_{c \in \mathcal{C}} \sum_{t=1}^{T} \langle\, \psi(c_t)\,,\, V\,\widehat{g}_t(x)\,\rangle_{\mathcal{H}} \qquad (29)$$

In particular, the following Lemma provides the characterization of $\widehat{f}$ for the case where the $\widehat{g}_t$ are the solution of kernel ridge regression applied independently for each task $t = 1, \ldots, T$ with training set $(x_{it}, \psi(y_{it}))_{i=1}^{n_t}$

$$\widehat{g}_t(x) = \operatorname*{argmin}_{g \in \mathcal{H} \otimes \mathcal{G}} \frac{1}{n_t}\sum_{i=1}^{n_t} \|\psi(y_{it}) - g(x_{it})\|^2 + \lambda\|g\|_{\mathcal{H} \otimes \mathcal{G}}^2$$

where $\mathcal{G}$ is a reproducing kernel Hilbert space with associated reproducing kernel $k : \mathcal{X} \times \mathcal{X} \to \mathbb{R}$. Recall that since $\mathcal{H}$ and $\mathcal{G}$ are reproducing kernel Hilbert spaces, also $\mathcal{H} \otimes \mathcal{G}$ is.

**Proposition 3** (The Nonlinear MTL Estimator). *Let $k : \mathcal{X} \times \mathcal{X} \to \mathbb{R}$ be a reproducing kernel with associated reproducing kernel Hilbert space $\mathcal{G}$. Let $\widehat{g}_t : \mathcal{X} \to \mathcal{H}$ be the solution of Eq. (12) for $t = 1, \ldots, T$. Then the estimator $\widehat{f} : \mathcal{X} \to \mathcal{C}$ defined at Eq. (11) is such that*

$$\widehat{f}(x) = \operatorname*{argmin}_{c \in \mathcal{C}} \sum_{t=1}^{T}\sum_{i=1}^{n_t} \alpha_{it}(x)\ell(c_t, y_{it}), \quad (\alpha_{1t}(x), \ldots, \alpha_{n_t t}(x))^\top = (K_t + n_t\lambda_t I)^{-1}K_{tx} \quad (13)$$

*for all $x \in \mathcal{X}$ and $t = 1, \ldots, T$, where $K_t \in \mathbb{R}^{n_t \times n_t}$ denotes the kernel matrix of the $t$-th task, namely $(K_t)_{ij} = k(x_{it}, x_{jt})$, and $K_{tx} \in \mathbb{R}^{n_t}$ the vector with $i$-th component equal to $k(x, x_{it})$.*

*Proof.* First, note that the solution of the problem above has the form of

$$\widehat{g}_t(x) = \sum_{i=1}^{n_t} \alpha_{it}(x)\psi(y_{it})$$

with $\alpha_{it} : \mathcal{X} \to \mathbb{R}$ defined as in Eq. (13) (This is a standard result when $\mathcal{H}$ is finite dimensional, see [2] Lemma 17 and Eq.(88)-(89) for the infinite dimensional case. Then, according to the SELF characterization of the loss $\ell$ in terms of $\psi$ and $V$, we have for any $x \in \mathcal{X}$

$$\frac{1}{T}\sum_{t=1}^{T}\sum_{i=1}^{n_t} \alpha_{it}(x)\ell(c_t, y_{it}) = \frac{1}{T}\sum_{t=1}^{T}\sum_{i=1}^{n_t} \alpha_{it}(x)\langle \psi(c_t), V y_{it}\rangle_H \tag{30}$$

$$= \frac{1}{T}\sum_{t=1}^{T} \langle \psi(c_t), V \sum_{i=1}^{n_t} \alpha_{it}(x)y_{it}\rangle_H \tag{31}$$

$$= \frac{1}{T}\sum_{t=1}^{T} \langle \psi(c_t), V \widehat{g}_t(x)\rangle_H. \tag{32}$$

$\square$

In the following section we study the generalization properties of such estimator as the number of examples per task grows.

## A.2   A Comparison Inequality for Multitask Learning

In this section we prove the comparison inequality of Thm. 5, which is key to study the generalization performance of nonlinear MTL.

**Theorem 5** (Comparison Inequality). *With the same assumptions of Prop. 2, for $t = 1, \ldots, T$ let $f^* : \mathcal{X} \to \mathcal{C}$ and $g_t^* : \mathcal{X} \to \mathcal{H}$ be defined as in Eq. (10), let $\widehat{g}_t : \mathcal{X} \to \mathcal{H}$ be measurable functions and let $\widehat{f} : \mathcal{X} \to \mathcal{C}$ satisfy Eq. (11). Let $V^*$ be the adjoint of $V$. Then,*

$$\mathcal{E}(\widehat{f}) - \mathcal{E}(f^*) \leq q_{\mathcal{C},\ell,T} \sqrt{\frac{1}{T}\sum_{t=1}^{T} \|\widehat{g}_t - g_t^*\|_{L_{\rho_\mathcal{X}}^2}^2}, \quad q_{\mathcal{C},\ell,T} = 2\sup_{c\in\mathcal{C}} \sqrt{\frac{1}{T}\sum_{t=1}^{T} \|V^*\psi(c_t)\|_{\mathcal{H}}^2}. \tag{18}$$

*Proof.* Recall that for any measurable $g_t : \mathcal{X} \to \mathcal{H}$,

$$\|g_t^* - g_t\|_{L^2(\mathcal{X},\mathcal{H},\rho)}^2 := \int \|g_t^*(x) - g_t(x)\|_{\mathcal{H}}^2 d\rho(x).$$

Let $\bar{\mathcal{E}}$ be defined as

$$\bar{\mathcal{E}}(f) = \int_{\mathcal{X}} \frac{1}{T}\sum_{t=1}^{T} \langle \psi(f(x)_t), V \widehat{g}_t(x)\rangle_{\mathcal{H}_\mathcal{Y}} d\rho(x),$$

for any measurable function $f : \mathcal{X} \to \mathcal{C}$. We have

$$\mathcal{E}(f) - \mathcal{E}(f^*) = \mathcal{E}(f) - \bar{\mathcal{E}}(f) + \bar{\mathcal{E}}(f) - \mathcal{E}(f^*) = A + B \tag{33}$$

We will bound separately the terms $A$ and $B$. By the characterization of $\mathcal{E}$ in Lemma 9, we have

$$A = \mathcal{E}(f) - \bar{\mathcal{E}}(f) = \frac{1}{T} \int_{\mathcal{X}} \sum_{t=1}^{T} \langle \psi(f(x)_t), V(g_t^*(x) - \widehat{g}_t(x)) \rangle_{\mathcal{H}} d\rho(x) \tag{34}$$

$$\leq \frac{1}{T} \int_{\mathcal{X}} \sum_{t=1}^{T} \|V^*\psi(f(x)_t)\|_{\mathcal{H}} \|g_t^*(x) - \widehat{g}_t(x)\|_{\mathcal{H}} d\rho(x) \tag{35}$$

$$\leq \frac{1}{T} \sum_{t=1}^{T} \sup_{c \in \mathcal{C}} \|V^*\psi(c_t)\|_{\mathcal{H}} \int_{\mathcal{X}} \|g_t^*(x) - \widehat{g}_t(x)\|_{\mathcal{H}} d\rho(x) \tag{36}$$

$$= \frac{1}{T} \sum_{t=1}^{T} \sup_{c \in \mathcal{C}} \|V^*\psi(c_t)\|_{\mathcal{H}} \sqrt{\int_{\mathcal{X}} \|g_t^*(x) - \widehat{g}_t(x)\|_{\mathcal{H}}^2 d\rho(x)} \tag{37}$$

$$= \frac{1}{T} \sqrt{\sum_{t=1}^{T} \sup_{c \in \mathcal{C}} \|V^*\psi(c_t)\|_{\mathcal{H}}^2} \sqrt{\sum_{t=1}^{T} \int_{\mathcal{X}} \|g_t^*(x) - \widehat{g}_t(x)\|_{\mathcal{H}}^2 d\rho(x)}. \tag{38}$$

Since, by Lemma 10 $f^*$ is a minimizer of $\mathcal{E}$ and with the same reasoning $f$ is a minimizer of $\bar{\mathcal{E}}$, we have

$$B = \bar{\mathcal{E}}(f) - \mathcal{E}(f^*) \tag{39}$$

$$= \int_{\mathcal{X}} \min_{c \in \mathcal{C}} \frac{1}{T} \sum_{t=1}^{T} \langle \psi(c_t), V\widehat{g}_t(x) \rangle_{\mathcal{H}} \, d\rho(x) \;-\; \int \min_{c \in \mathcal{C}} \frac{1}{T} \sum_{t=1}^{T} \langle \psi(c_t), V g_t^*(x) \rangle_{\mathcal{H}} \, d\rho(x) \tag{40}$$

$$= \frac{1}{T} \int_{\mathcal{X}} \min_{c \in \mathcal{C}} \sum_{t=1}^{T} \langle \psi(c_t), V\widehat{g}_t(x) \rangle_{\mathcal{H}} \;-\; \min_{c \in \mathcal{C}} \sum_{t=1}^{T} \langle \psi(c_t), V g_t^*(x) \rangle_{\mathcal{H}} \, d\rho(x) \tag{41}$$

$$\leq \frac{1}{T} \int_{\mathcal{X}} \sup_{c \in \mathcal{C}} \left| \sum_{t=1}^{T} \langle \psi(c_t), V\widehat{g}_t(x) \rangle_{\mathcal{H}_{\mathcal{Y}}} - \sum_{t=1}^{T} \langle \psi(c_t), V g_t^*(x) \rangle_{\mathcal{H}} \right| \, d\rho(x) \tag{42}$$

$$= \frac{1}{T} \int_{\mathcal{X}} \sup_{c \in \mathcal{C}} \left| \sum_{t=1}^{T} \langle V^*\psi(c_t), (\widehat{g}_t(x) - g_t^*(x)) \rangle_{\mathcal{H}} \right| \, d\rho(x) \tag{43}$$

$$\leq \frac{1}{T} \int_{\mathcal{X}} \sup_{c \in \mathcal{C}} \sum_{t=1}^{T} \|V^*\psi(c_t)\|_{\mathcal{H}} \|\widehat{g}_t(x) - g_t^*(x)\|_{\mathcal{H}} \, d\rho(x) \tag{44}$$

$$\leq \frac{1}{T} \int_{\mathcal{X}} \sup_{c \in \mathcal{C}} \sqrt{\sum_{t=1}^{T} \|V^*\psi(c_t)\|_{\mathcal{H}}^2} \sqrt{\sum_{t=1}^{T} \|\widehat{g}_t(x) - g_t^*(x)\|_{\mathcal{H}}^2} \, d\rho(x) \tag{45}$$

$$\leq \frac{1}{T} \sqrt{\sup_{c \in \mathcal{C}} \sum_{t=1}^{T} \|V^*\psi(c_t)\|_{\mathcal{H}}^2} \sqrt{\int_{\mathcal{X}} \sum_{t=1}^{T} \|\widehat{g}_t(x) - g_t^*(x)\|_{\mathcal{H}}^2 d\rho(x)}. \tag{46}$$

as desired. $\qquad\square$

### A.3 Consistency and Generalization Bounds

The comparison inequality provided by Thm. 5 allows to characterize the generalization properties of the estimator $\widehat{f} : \mathcal{X} \to \mathcal{C}$ by studying the functions $\widehat{g}_t : \mathcal{X} \to \mathcal{H}$. In particular, if $\widehat{g}_t \to g_t^*$ in $L_2$ for all $t = 1, \ldots, T$, the comparison inequality automatically guarantees that the excess risk $\mathcal{E}(\widehat{f}) - \mathcal{E}(f^*) \to 0$. Moreover, if we are able to determine the rate for which the $\widehat{g}_t$ converge to the $g_t^*$, we can provide generalization bounds for the SELF estimator $\widehat{f}$.

The following results formalize these ideas, proving consistency and learning rates for $\widehat{f}$.

**Theorem 6.** *Let $\mathcal{C} \subseteq [a, b]^T$, let $\mathcal{X}$ be a compact set and $k : \mathcal{X} \times \mathcal{X} \to \mathbb{R}$ a continuous universal reproducing kernel (e.g. Gaussian). Let $\ell : [a, b] \times [a, b] \to \mathbb{R}$ be a SELF. Let $\widehat{f}_N : \mathcal{X} \to \mathcal{C}$ denote*

the estimator at Eq. (13) with $N = (n_1, \ldots, n_T)$ training points independently sampled from $\rho_t$ for each task $t = 1, \ldots, T$ and $\lambda_t = n_t^{-1/4}$. Let $n_0 = \min_{1 \leq t \leq T} n_t$. Then, with probability 1

$$\lim_{n_0 \to +\infty} \mathcal{E}(\widehat{f}_N) = \inf_{f:\mathcal{X} \to \mathcal{C}} \mathcal{E}(f). \tag{19}$$

*Proof.* By the comparison inequality in Thm. 5 it is sufficient to prove that $\widehat{g}_t \to g_t^*$ in $L^2$ in probability as the number of training point increases. However note that $g_t^* : \mathcal{X} \to \mathcal{H}$ such that $g_t^*(x) = \int_{\mathbb{R}} \psi(y) \, d\rho_t(y|x)$ is the minimizer of the least squares expected risk

$$\underset{g:\mathcal{X} \to \mathcal{H}}{\text{minimize}} \quad \int_{\mathcal{X} \times \mathcal{Y}} \|g(x) - \psi(y)\|_{\mathcal{H}}^2 \, d\rho_t(x, y). \tag{47}$$

Indeed,

$$\int_{\mathcal{X} \times \mathcal{Y}} \|g(x) - \psi(y)\|_{\mathcal{H}}^2 \, d\rho_t(x, y) = \int_{\mathcal{X} \times \mathcal{Y}} \|g(x)\|_{\mathcal{H}}^2 - 2\langle g(x), \psi(y) \rangle \, d\rho_t(x, y) + const$$

$$= \int_{\mathcal{X}} \|g(x)\|_{\mathcal{H}}^2 - 2\langle g(x), \int_{\mathcal{Y}} \psi(y) \, d\rho_t(y|x) \rangle \, d\rho_{\mathcal{X}}(x) + const$$

$$= \int_{\mathcal{X}} \|g(x)\|_{\mathcal{H}}^2 - 2\langle g(x), g_t^*(x) \rangle \, d\rho_{\mathcal{X}}(x) + const$$

Which is clearly minimized pointwise for each $x \in \mathcal{X}$ by $g(x) = g_t^*(x)$.

Therefore, in order to study the convergence of the kernel ridge regression estimator $\hat{g}_t$ to $g_t^*$ we can leverage on either standard results for kernel ridge regression (e.g. see [3]) if $\mathcal{H}$ is finite dimensional or the result in [2] (Lemma 18 see Eq. 96) when $\mathcal{H}$ is infinite dimensional. Both analyses provide analogous bounds on $\|\widehat{g}_t - g_t^*\|_{L^2}$ with respect to the number of training examples $n_t$. In particular, the direct application of [2] (Lemma 18) to our setting, leads to the desired convergence of $\widehat{g}_t$ to $g_t^*$, and consequently of $\mathcal{E}(\widehat{f})$ to $\mathcal{E}(f^*)$ as desired. $\qquad \square$

**Theorem 7.** *With the same assumptions and notation of Thm. 6 let $\widehat{f}_N : \mathcal{X} \to \mathcal{C}$ denote the estimator at Eq. (13) with $\lambda_t = n_t^{-1/2}$ and assume $g_t^* \in \mathcal{H} \otimes \mathcal{G}$, for all $t = 1, \ldots, T$. Then for any $\tau > 0$ we have, with probability at least $1 - 8e^{-\tau}$, that*

$$\mathcal{E}(\widehat{f}_N) - \inf_{f:\mathcal{X} \to \mathcal{C}} \mathcal{E}(f) \leq q_{\mathcal{C}, \ell, T} \ h_\ell \ \tau^2 \ n_0^{-1/4} \log T, \tag{20}$$

*where $q_{\mathcal{C}, \ell, T}$ is defined as in Eq. (18) and $h_\ell$ is a constant independent of $\mathcal{C}, N, n_t, \lambda_t, \tau, T$.*

*Proof.* The same reasoning in the proof of Thm. 6 applies. In particular, since $g_t^* \in \mathcal{H} \otimes \mathcal{G}$ for every $t = 1, \ldots, T$, we can improve the bound in [2, Lemma 18] analogously to what is done in [2, Thm. 5], leading, for any $\eta > 0$, to the bound

$$\|g_t - g_t^*\|_{L^2} \leq h_{\ell, t} \ \eta^2 \ n_t^{-1/4}$$

holding with probability at least $1 - 8e^{-\eta}$, where

$$h_{\ell, t} = 12(\Psi + \kappa \|g_t^*\|_{\mathcal{H} \otimes \mathcal{G}} + \kappa) \qquad \text{with} \qquad \Psi = \sup_{y \in \mathcal{Y} = [a, b]} \|\psi(y)\|_{\mathcal{H}} \text{ and } \kappa = \sqrt{\sup_{x \in \mathcal{X}} k(x, x)}$$

is a constant not depending on $\mathcal{C}, N, n_t, \eta, \lambda_t$. By taking the intersection bound of these events, we have that the following holds with probability $1 - T 8 e^{-\eta}$

$$\sqrt{\frac{1}{T} \sum_{t=1}^{T} \|g_t - g_t^*\|_{L^2}^2} \leq \sqrt{\frac{1}{T} \sum_{t=1}^{T} h_{\ell, t}^2 \ \eta^4 \ n_t^{-1/2}} \leq h_\ell \ \eta^2 \ n_0^{-1/4}.$$

with $h_\ell = \max_{t=1,\ldots,T} h_{\ell, t}$. Then, by choosing $\eta := \tau \log(T)$ we have

$$\sqrt{\frac{1}{T} \sum_{t=1}^{T} \|g_t - g_t^*\|_{L^2}^2} \leq h_\ell \ \tau^2 \ n_0^{-1/4} \log(T),$$

with probability $1 - 8e^{-\tau}$. Combining the equation above with the comparison inequality we have the desired generalization bound. $\qquad \square$

We conclude proving the result studying the constant $q_{\mathcal{C},\ell,T}$ in the case of $\mathcal{C} = [-B, B]^2$ and $\mathcal{C} = \mathcal{S}_T$ the sphere of radius B.

**Lemma 8.** *Let $B \geq 1$, $\mathcal{B} = [-B, B]^T$, $\mathcal{S} \subset \mathbb{R}^T$ be the sphere of radius $B$ centered at the origin and let $\ell$ be the square loss. Then $q_{\mathcal{B},\ell,T} \leq 2\sqrt{5} \, B^2$ and $q_{\mathcal{S},\ell,T} \leq 2\sqrt{5} \, B^2 \, T^{-1/2}$.*

*Proof.* We begin by observing that the least square expected risk can be equivalently written as

$$\mathcal{E}(f) = \min_{f:\mathcal{X}\to\mathcal{C}} \frac{1}{T} \sum_{t=1}^{T} \int_{\mathcal{X}\times\mathcal{Y}} (f(x) - y)^2 \; d\rho_t(x, y) \tag{48}$$

$$= \min_{f:\mathcal{X}\to\mathcal{C}} \frac{1}{T} \sum_{t=1}^{T} \int_{\mathcal{X}\times\mathcal{Y}} f(x)^2 - f(x)y + y^2 \; d\rho_t(x, y) \tag{49}$$

$$= \min_{f:\mathcal{X}\to\mathcal{C}} \frac{1}{T} \sum_{t=1}^{T} \int_{\mathcal{X}\times\mathcal{Y}} f(x)^2 - 2f(x)y \; d\rho_t(x, y). \tag{50}$$

Therefore we can equivalently study the asymmetric loss $\ell : \mathbb{R}\times\mathbb{R} \to \mathbb{R}$ such that $\ell(y, y') = y^2 - 2yy'$. Such loss can be written in SELF form $\ell(y, y') = \langle \psi(y), V\psi(y')\rangle_{\mathcal{H}}$ with $\mathcal{H} = \mathbb{R}^3$,

$$\psi(y) = (y^2, y, 1)^\top \in \mathbb{R}^2 \qquad \text{and} \qquad V = \begin{pmatrix} 0 & 0 & 1 \\ 0 & -2 & 0 \\ 1 & 0 & 0 \end{pmatrix}.$$

Therefore we have $\|V^*\psi(y)\|^2 = 4y^2 + y^4$ for all $y \in \mathbb{R}$. In particular we have that for every $c = (c_1, \ldots, c_T)^\top \in \mathcal{C} \subseteq \mathbb{R}$

$$\sqrt{\frac{1}{T} \sum_{t=1}^{T} \|V^*\psi(c_t)\|^2_{\mathcal{H}}} = \sqrt{\frac{1}{T} \sum_{t=1}^{T} 4c_t^2 + c_t^4}$$

Now we can provide the value for the constant $q_{\mathcal{C},\ell,T}$ for the two case considered in this work.

**Case $\mathcal{C} = \mathcal{B} = [-B, B]^T$.** The supremum is achieved at the edges of the cube $\mathcal{B}$, e.g. $c = B(1, \ldots, 1)^\top \in \mathbb{R}^T$, namely

$$q_{\mathcal{B},\ell,T} = 2 \sup_{c \in \mathcal{C}} \sqrt{\frac{1}{T} \sum_{t=1}^{T} \|V^*\psi(c_t)\|^2_{\mathcal{H}}} = 2\sqrt{4B^2 + B^4} \leq 2\sqrt{5} \, B^2$$

**Case $\mathcal{C} = \mathcal{S}_{B,T}$.** the sphere of radius $B$ in $\mathbb{R}^T$ centered in zero. Since $\|c\| = B$ for any $c \in \mathcal{S}_B$ we have

$$q_{\mathcal{S}_{B,T},\ell} = 2\sqrt{\frac{4B^2 + B^4}{T}}$$

Since $c_t^2 \leq B^2$ for all $c \in \mathcal{S}_{B,T}$ and $t = 1, \ldots, T$. Therefore $\sum_{t=1}^{T} c_t^4 \leq B^2 \sum_{t=1}^{T} c_t = B^4$. However such value is achieved for $c = (B, 0, \ldots, 0)^\top \in \mathbb{R}^T$, hence $q_{\mathcal{C},\ell,T} = B\sqrt{4 + B^2}$. We conclude

$$q(\mathcal{S}_{B,T}, \ell) \leq 2\sqrt{5} \, B^2 \, T^{-1/2}$$

as desired. □

## B (Most) MTL Loss Functions are SELF

We conclude with a note on the results reported in Sec. 4 characterizing sufficient conditions for the SELF property to be satisfied by either mono-variate loss functions or a separable loss sum of SELF functions.

**Proposition 4.** *Let $\bar{\ell} : [a, b] \to \mathbb{R}$ be differentiable almost everywhere with derivative Lipschitz continuous almost everywhere. Let $\ell : [a, b] \times [a, b] \to \mathbb{R}$ be such that $\ell(y, y') = \bar{\ell}(y - y')$ or $\ell(y, y') = \bar{\ell}(yy')$ for all $y, y' \in \mathbb{R}$. Then: (i) $\ell$ is SELF and (ii) the separable function $\mathcal{L} : \mathcal{Y}^T \times \mathcal{Y}^T \to \mathbb{R}$ such that $\mathcal{L}(y, y') = \frac{1}{T} \sum_{t=1}^{T} \ell(y_t, y'_t)$ for all $y, y' \in \mathcal{Y}^T$ is SELF.*

*Proof.* The result is a corollary of Thm. 19 in [2]. In particular $(i)$ is a direct application of Thm. 19 point 2, while $(ii)$ follows from the combination of Thm. 19 point 4, namely the fact that we are studing the separable loss on a compact subset of $\mathbb{R}^T$ and point 6, implying that the sum of SELF functions is indeed SELF. □

## C  Nonlinear Multitask learning and Square Loss

We observed that for the square loss it is possible to derive a more compact and interpretable formulation for the solutions of the algorithms considered in this work (see Example 1 and Example 3). Here we show how these solution are derived in detail. In particular, in the notation of Example 3 denote the solution $\widehat{f}_0 : \mathcal{X} \to \mathcal{C}$ to the *exact* MTL problem with nonlinear constraints $\mathcal{C}$. $f_0$ needs to satisfy Eq. (8), where, for $\mathcal{L}(y, y') = \|y = y'\|^2$ the weighted sum $\sum_{i=1}^n \alpha_i(x) \mathcal{L}(y, y_i)$ is

$$\sum_{i=1}^n \alpha_i(x)\|c - y_i\|^2 = a(x)\|c\|^2 - 2c^\top b(x) + const \tag{51}$$

with $a(x) = \sum_{i=1}^n \alpha_i(x)$ and $b(x) = \sum_{i=1}^n \alpha_i(x)y_i$. Therefore $f_0$ corresponds to the projection

$$\widehat{f}_0(x) = \underset{c \in \mathcal{C}}{\operatorname{argmin}} \ \|c - b(x)/a(x)\|^2 = \Pi_{\mathcal{C}}(b(x)/a(x)) \tag{52}$$

**Solution for the Robust $C_\delta$ in Example 3**. The algorithm with constraint set $\mathcal{C}_\delta$ takes the form

$$\widehat{f}_\delta(x) = \underset{c \in \mathcal{C}, \|r\| \leq \delta}{\operatorname{argmin}} \ \sum_{i=1}^n \alpha_i(x)\|c + r - y_i\|^2. \tag{53}$$

Following the same reasoning as in Example 1, we have that this problem is equivalent to

$$\widehat{f}_\delta(x) = \underset{c \in \mathcal{C}, \|r\| \leq \delta}{\operatorname{argmin}} \ \|c + r - b(x)/a(x)\|^2. \tag{54}$$

By solving Eq. (54) in $r$ we have

$$r(c) = \underset{\|r\| \leq \delta}{\operatorname{argmin}} \|r - (b(x)/a(x) - c)\|^2 \tag{55}$$

which is solved by

$$r(c) = (b(x)/a(x) - c) \min(1, \frac{\delta}{\|b(x)/a(x) - c\|}). \tag{56}$$

Substituting $r(c)$ in Eq. (54), we have

$$c_\delta = \underset{c \in \mathcal{C}}{\operatorname{argmin}} \|c + r(c) - b(x)/a(x)\|^2 \tag{57}$$

$$= \underset{c \in \mathcal{C}}{\operatorname{argmin}} \|c - b(x)/a(x)\|^2 (1 - \min(1, \delta/\|c - b(x)/a(x)\|))^2 \tag{58}$$

which is minimized for $c_\delta = \Pi_{\mathcal{C}}(b(x)/a(x)) = \widehat{f}_0(x)$. Therefore, $\widehat{f}_\delta(x)$ is

$$\widehat{f}_\delta(x) = c_\delta + r_\delta = \widehat{f}_0(x) + (b(x)/a(x) - \widehat{f}_0(x)) \min(1, \frac{\delta}{\|b(x)/a(x) - \widehat{f}_0(x)\|}) \tag{59}$$

as stated in Example 3.

**Solution for Perturbed loss $\mathcal{L}_\mu$ in Example 3**. For simplicity of notation denote $\delta = 1/\mu$. We have that the vector-valued learning algorithm with perturbed loss $\mathcal{L}_{1/\delta}$ is

$$\widehat{f}_{1/\delta}(x) = \underset{c \in \mathcal{C}, r \in \mathbb{R}^M}{\operatorname{argmin}} \ \sum_{i=1}^n \alpha_i(x)(\|c + r - y_i\|^2 + \delta\|r\|^2). \tag{60}$$

By deriving the functional w.r.t. $r$ and setting it to zero we have the equation,

$$\sum_{i=1}^n \alpha_i(x)((c + r - y_i) + \delta r) = 0 \tag{61}$$

which implies that the minimizer $r(c) \in \mathbb{R}^M$ of Eq. (60) for any $c \in \mathcal{C}$ fixed, is

$$r(c) = \frac{b(x) - a(x)c}{(1 + \delta)a(x)} \in \mathbb{R}^M. \tag{62}$$

Now, plugging $r(c)$ into $\sum_{i=1}^{n} \alpha_i(x)\|c + r - y_i\|^2$, we have

$$\sum_{i=1}^{n} \alpha_i(x)\|c + r(c) - y_i\|^2 = \sum_{i=1}^{n} \alpha_i(x)\|\frac{\delta c}{(1 + \delta)} - y_i + \frac{b(x)}{(1 + \delta)a(x)}\|^2 \tag{63}$$

$$= \frac{\delta^2 a(x)}{(1 + \delta)^2}\|c\|^2 - 2\frac{\delta}{(1 + \delta)}c^\top \left(\sum_{i=1}^{n} \alpha_i(x)y_i - \frac{b(x)}{(1 + \delta)}\right) + \text{const} \tag{64}$$

$$= \frac{\delta^2}{(1 + \delta)^2}\left(a(x)\|c\|^2 - 2c^\top b(x)\right) + \text{const} \tag{65}$$

where we have used the fact that $b(x) = \sum_{i=1}^{n} \alpha_i(x)y_i$ and denoted with const every addend not depending on $c$.

We now insert $r(c)$ in $\sum_{i=1}^{n} \alpha_i(x)\delta\|r\|^2$ and obtain

$$\sum_{i=1}^{n} \alpha_i(x)\delta\|r\|^2 = \frac{a(x)\delta}{(1 + \delta)^2 a(x)^2}\|b(x) - a(x)c\|^2 \tag{66}$$

$$= \frac{\delta}{(1 + \delta)^2 a(x)}\left(a(x)^2\|c\|^2 - 2a(x)c^\top b(x)\right) + \text{const} \tag{67}$$

$$= \frac{\delta}{(1 + \delta)^2}(a(x)\|c\|^2 - 2c^\top b(x)) + \text{const}. \tag{68}$$

We can therefore plug $r(c)$ into Eq. (60), obtaining

$$c_{1/\delta} = \underset{c \in \mathcal{C}}{\text{argmin}} \sum_{i=1}^{n} \alpha_i(x)(\|c + r(c) - y\|^2 + \delta\|r(c)\|^2) \tag{69}$$

$$= \underset{c \in \mathcal{C}}{\text{argmin}} \frac{\delta}{(1 + \delta)}\left(a(x)\|c\|^2 - 2c^\top b(x)\right) \tag{70}$$

$$\tag{71}$$

which is minimized again for $c_{1/\delta} = \Pi_{\mathcal{C}}(b(x)/a(x)) =: \widehat{f}_0(x)$. By evaluating $r_{1/\delta} = r(c_{1/\delta})$ we have

$$r_{1/\delta}(x) = \frac{b(x)/a(x) - \widehat{f}_0(x)}{1 + \delta} \tag{72}$$

and $\widehat{f}_{1/\delta}(x) = \widehat{f}_0(x) + r_{1/\delta}(x)$. Finally, by taking $\mu = 1/\delta$ we recover the solution at Example 3.

We conclude by reporting a closed form solution for the MTL estimator with perturbed loss functions $\ell_\mu^t$ for each task. The derivation is analogous to that for the vector-valued case.

**Example 4** (Nonlinear MLT and Violations). *With the same notation as in Example 2 let us define* $\bar{a}(x) = \sum_{t=1}^{T} \sum_{i=1}^{n_t} \alpha_{it}(x)$. *Denote* $\widehat{f}_0 : \mathcal{X} \to \mathcal{C}$ *the map such that for all* $x \in \mathcal{X}$

$$\widehat{f}_0(x) = \underset{c \in \mathcal{C}}{\text{argmin}} \sum_{t=1}^{T} \frac{a_t(x)}{a_t(x) + \bar{a}(x)/\mu}\left(c_t - \frac{b_t(x)}{a_t(x)}\right)^2 \tag{73}$$

*which corresponds to the projection of the vector* $(b_1(x)/a_1(x), \ldots, b_T(x)/a_T(x))$ *onto* $\mathcal{C}$ *according to the diagonl matrix* $\Sigma_x$ *with diagonal elements* $a_t(x)/(a_t(x) + \bar{a}(x)/\mu)$ *for* $t = 1, \ldots, T$.

*The solution with perturbed loss functions* $\ell_\mu^t$ *is* $f_\mu : \mathcal{X} \to \mathbb{R}^T$ *such that for all* $x \in \mathcal{X}$, *the* $t - th$ *coordinate of* $\widehat{f}_\mu(x)$ *is*

$$\widehat{f}_\mu(x)_t = \frac{b_t(x)/a_t(x) + \bar{a}(x)/\mu \, \widehat{f}_0(x)}{a_t(x) + \bar{a}(x)/\mu}. \tag{74}$$

# D   Experiments

**Synthetic Dataset**. We considered a model of the form $y = f^*(x) + \epsilon$ with $f^* : \mathcal{X} \to \mathcal{C}$ and $\epsilon \sim \mathbb{N}(0, \sigma I)$ noise sampled according to a normal isotropic distribution. We considered the case of two tasks constrained on $\mathcal{C} \subset \mathbb{R}^2$, either a circumference or a lemniscate (see Fig. 1), identified by the equations $\gamma_{\text{circ}}(y) = y_1^2 + y_2^2 - 1 = 0$ and $\gamma_{\text{lemn}}(y) = y_1^4 - (y_1^2 - y_2^2) = 0$ with $y = (y_1, y_2) \in \mathbb{R}^2$. We set $\mathcal{X} = [-\pi, \pi]$ and chose $f^* : [-\pi, \pi] \to \mathbb{R}^2$ to be parametric functions associated to the circumference and Lemniscate respectively, namely $f^*_{\text{circ}}(x) = (\cos(x), \; \sin(x))$ and $f^*_{\text{lemn}}(x) = (\sin(x), \; \sin(2x))$. We sampled from 10 to 1000 points for training and 1000 for testing, with noise $\sigma = 0.05$.

We trained and tested different models over 10 trials. We used a Gaussian kernel on the input and chose the corresponding bandwidth and the regularization parameter $\lambda$ by hold-out cross-validation on 30% of the training set for 30 values of the two hyperparameters sampled respectively from the ranges $[0.01, 100]$ and $[1e - 9, 1]$ with logarithmic spacing. Fig. 1 (Bottom) reports the mean square error (MSE) of the nonlinear MTL approach (NL-MTL) compared with the standard least squares single task learning (STL) baseline and the multitask relations learning (MTRL) from [4], which encourages tasks to be linearly dependent. For both the circumference and Lemniscate, the tasks are strongly *nonlinearly* related. As a consequence our approach consistently outperforms its two competitors which assume only linear relations (or none at all). Fig. 1 (Top) provides a qualitative comparison on the three methods (when trained with 100 examples) during a single trial.

**Sarcos Dataset**. We next report experiments on the Sarcos dataset [5]. The goal is to predict the torque measured at each joint of a 7 degrees-of-freedom robotic arm, given the current state, velocities and accelerations measured at each joint (7 tasks/torques for 21-dimensional input). We used the 10 dataset splits available online for the dataset in [6], each containing 2000 examples per task with 15 examples used for training/validation while the rest is used to measure errors in terms of the *explained variance*, namely 1 - nMSE (as a percentage). In order to be fair with results in [6] we used the canonical linear kernel on the input. We chose the hyperparamters each from 30 values sampled in the ranges $[1e^{-5}, 1e^5]$ for $\delta$ and $\mu$ and $[1e^{-9}, 1]$ for $\lambda$ with logarithmic spacing.

Tab. 1 reports results from [6, 7] for a wide range of previous *linear* MTL methods [8, 9, 10, 4, 6, 7], together with our nonlinear MTL approach (both the robust and perturbed versions). Since, we were not able to find the model parameters of the Sarcos robot online, we approximated the constraint set $\mathcal{C}$ as a point cloud by collecting 1000 random output vectors that neither belonged to the training nor to the test sets by sampling them from the original sarcos dataset [5]. As can be noticed, the approach proposed in this paper clearly outperforms the "linear" competitors. Note that indeed, the relations among the torques measured at different joints of a robot are highly nonlinear (see for instance [11]) and therefore taking such structure into account can be beneficial to the learning process.

**Ranking by Pair-wise Comparison.** We considered a ranking prediction problem, formulated within a non-linear multi-task learning setting. In particular, given a database of $D$ documents, the goal is to learn $T = D(D-1)/2$ functions $f_{p,q} : \mathcal{X} \to \{-1, 0, 1\}$, one for each pair of documents $p$ and $q$ from 1 to $D$, predicting if one document is more relevant than the other for a given input query $x$. Recommender systems are a natural application in this setting, and here we discuss an experiment on the Movielens100k [12] dataset for movie recommendation, where movies correspond to documents and queries to users.

The problem can be tackled as multi-label MTL with 0-1 loss: for a given training query $x$ only some pairwise comparisons $y_{p,q} \in \{-1, 0, 1\}$ are available (with 1 corresponding to movie $p$ being more relevant than $q$, $-1$ the opposite and 0 that the two movies are equivalent). Errors are measured according to the loss function

$$\triangle_{pairwise}(c, y) = \sum_{p=1}^{D-1} \sum_{q=p+1}^{D} \mathbf{1}_{\{c_{p,q} \neq y_{p,q}\}} \tag{75}$$

with $\mathbf{1}_{\{c_{p,q} \neq y_{p,q}\}} = 1$ if $c_{p,q} \neq y_{p,q}$ and zero otherwise. We have therefore $T$ separate training sets, one for each task (i.e. pair of movies). Clearly, not all possible combinations of outputs $f : \mathcal{X} \to \{-1, 0, 1\}^T$ are allowed: the comparisons need to be logically consistent (e.g. if $p \succ q$ (read "$p$ more relevant than $q$") and $q \succ r$, then we cannot have $r \succ p$). As shown in [13] these constraints are naturally encoded in a set $\mathcal{C} = DAG(D)$ in $\mathbb{R}^T$ of all vectors $G \in \mathbb{R}^T$ that correspond to (the vectorized, upper triangular part of the adjacency matrix of) a Directed Acyclic Graph with $D$

vertices. This leads to the NL-MTL estimator,

$$\hat{f}(x) = \operatorname*{argmin}_{c \in DAG(D)} \sum_{i=1}^{n} \sum_{p=1}^{D-1} \sum_{q=p+1}^{D} \alpha_{i,(p,q)}(x) \, \mathbf{1}_{\{c_{p,q} \neq y_{p,q}\}} \tag{76}$$

where each score function $\alpha_{i,(p,q)}(x)$ is learned according to Eq. (13) using only the training inputs for which the pairwise comparison $y_{p,q}$ is available. As already observed in [2], the above optimization over $DAG(D)$ can be formulated as a *minimal feedback Arc Set* problem on graphs [14] that can be addressed using approximation methods [15].

As a final remark, note that in the ranking prediction settings of [13, 2], the authors considered a *weighted* version of the binary loss at Eq. (75) of the form

$$\triangle_{rank}(c, y) = \sum_{p=1}^{D-1} \sum_{q=p+1}^{D} |y_{p,q}| \, \mathbf{1}_{\{c_{p,q} \neq \operatorname{sign}(y_{p,q})\}}$$

where $y_{p,q} \in \mathbb{R}$ is a scalar value measuring the discrepancy between documents $q$ and $q$ and is used to weight misclassification errors according to the relative importance between documents $p$ and $q$. Indeed, in the Movielens100k dataset, each movie $p$ was assigned a rating by the user (a discrete value $r_p$ from 1 to 5) and in [13] the weights in $\triangle_{rank}$ were chosen as $y_{p,q} = r_p - r_q$. It is clear that this variant of $\triangle_{pairwise}$ leads to an NL-MTL estimator analogous to the one in Eq. (76), for which the optimization over $\mathcal{C} = DAG(D)$ can still be tackled with minimal feedback arc set solvers. In particular for our experiments we used the publicly avaialable igraph library[2] which implements a number of standard algorithms to solve problems on graphs.

We performed experiments on Movielens100k to compare our MTL estimator with both standard multi-task learning baseline as well as with methods specifically designed to address ranking problem. In particular, we used the linear kernel and the train, validation and test splits adopted in [2] to perform 10 independent trials with 5-fold cross-validation for model selection. In Tab. 2 we report the average ranking error and standard deviation of the weighed 0-1 loss function considered in [13, 2] for the ranking methods proposed in [16, 17, 13], the SVMStruct estimator [18], the SELF estimator considered in [2] for ranking, the MTRL predictor and the STL baseline, corresponding to individual SVMs trained for each pairwise comparison. Results for previous methods are reported from [2].

As can be observed, NL-MTL outperforms all competitors, achieving better performance than the the original SELF estimator. Indeed it can be shown that [2] tackles pairwise-based ranking in a vector-valued fashion, by filling missing observations with $0$. This has a negative effect, artificially biasing predictions towards $0$. Conversely, NL-MLT exploits the true MTL nature of the problem leading to superior performance. As expected, STL and MTRL are clearly not suited for this setting, since they miss the critical information contained in the constraint set.