[Reviews · NeurIPS 2017]

Reviewer 1



The paper tackles multi-task learning problems where there are non-linear relationships between tasks. The relationships between tasks is encoded as a set of non-linear constraints that the outputs of each task must satisfy (e.g . y1^2 + y2^2 = 1). In a nutshell, he proposed technique can be summarized as: use kernel regression to make predictions for each task independently, then project the prediction vector onto the constrained set. Overall, I like the idea of being able to take advantage of non-linear relationships between tasks. However, I am not sure how practical it is to specify the non-linear constraints between tasks in practice. The weakest point of the paper is the empirical evaluation. It would be good if the authors can discuss real applications where one can specify a reasonable set non-linear relationships between tasks. The examples provided in the paper are not very satisfactory. One imposes deterministic relationships between tasks (i.e. if the output for task 1 is known then the output for task 2 can be inferred exactly), and the other approximates the relationship through a point cloud (which raises the question why was the data in the point cloud not used as training data. Also to obtain a point cloud one needs to be in a vector-valued regression setting). The paper could be significantly improved by providing a more thorough empirical evaluation section that analyses the performance of the proposed method on multiple real problems. One of the main contributions of the paper, as stated by the authors, is the extension of the SELF approach beyond vector-valued regression. But evaluation is done only on vector-valued regression problems. The technique depends on using a kernel K for the kernel regression. Is there any guidance on how to select such a kernel? What kernel was used in the experimental section? It would also be good if the authors analyze the effect of various design and parameter choices. How sensitive are the results to the kernel K, to the regularization parameter \lambda, to the parameters \delta and \mu? How were all the above parameters set in the experimental section? Looking at figure 1, I am very surprised that the performance of STL seems to converge to a point significantly worse than that of NL-MTL. Given a large amount of data I would expect STL to perform just as well or better than any MTL technique (information transfer between tasks is no longer relevant if there is enough training data). This does not seem to be the case in figure 1. Why is that?

Reviewer 2



Summary: The authors propose a method for multi-task learning with nonlinear output relations using kernel ridge regression combined with manifold/discrete optimization. They provide detailed theoretical analysis on the generalization properties of their method. Experiments show that the method works well on two synthetic data and a nonlinear dynamical systems dataset of a robot arm. Comments: - In general the paper is well-written and clear. I like the careful analysis on generalization bounds and convergence rate of the excess risk. That is a main contribution of the current paper. - The main difficulty of applying the current method is the identification of the constraint set C, which describes the nonlinear relationship between the tasks. For example, in the synthetic data the user needs to know the exact equation governing the curve, and in the SARCOS task extra examples are required to estimate the non-linear manifold of the outputs. If the constraint set C is mis-specified, which could be common in actual applications, the end result could be worse than STL, just like the mis-specified linear MTL models in the two synthetic datasets. There are many more ways to mis-specify a nonlinear relationship than a linear one. If there are not enough examples from the output space to estimate the joint relationship between tasks, the end result could be worse than not applying MTL.

Reviewer 3



This paper presents a formulation and algorithm for multi-task learning with nonlinear task relations. A set of constraints are required to be satisfied while optimizing the joint functions. Extensions are considered when the constraints cannot be all satisfied. The work is novel in the sense that most previous multitask works considered only linear relations among the tasks. An algorithms and synthetic as well as a robotic domain experiments are shown to evaluate the approach. The paper is theoretically sound and I find the algorithm interesting, although one question in my mind is whether using linear relations to approximate the nonlinear ones is already sufficient in many real world domains. While Table 1 shows significant improvements of the nonlinear MTL approach, it is not clear whether the nonlinear MTL models suffer from overfitting due to increased model complexity. Perhaps more experiments in different problem domains will convince the reviewer.